# Perception of biological motion by jumping spiders

**Massimo De Agrò**[1,2,3]*, **Daniela C. Rößler**[1,2], **Kris Kim**[1,2], **Paul S. Shamble**[1,2]

**1** John Harvard Distinguished Science Fellows Program, Harvard University, Cambridge, Massachusetts, United States of America, **2** Department of Organismic and Evolutionary Biology, Harvard University, Cambridge, Massachusetts, United States of America, **3** Department of Zoology, Regensburg University, Regensburg, Germany

* massimo.deagro@gmail.com

**Data Availability Statement:** All relevant data are within the paper and its Supporting Information files.

**Funding:** PSS was supported by the John Harvard Distinguished Science Fellows Program within the FAS Division of Science, Harvard University. The

## Abstract

The body of most creatures is composed of interconnected joints. During motion, the spatial location of these joints changes, but they must maintain their distances to one another, effectively moving semirigidly. This pattern, termed "biological motion" in the literature, can be used as a visual cue, enabling many animals (including humans) to distinguish animate from inanimate objects. Crucially, even artificially created scrambled stimuli, with no recognizable structure but that maintains semirigid movement patterns, are perceived as animated. However, to date, biological motion perception has only been reported in vertebrates. Due to their highly developed visual system and complex visual behaviors, we investigated the capability of jumping spiders to discriminate biological from nonbiological motion using point-light display stimuli. These kinds of stimuli maintain motion information while being devoid of structure. By constraining spiders on a spherical treadmill, we simultaneously presented 2 point-light displays with specific dynamic traits and registered their preference by observing which pattern they turned toward. Spiders clearly demonstrated the ability to discriminate between biological motion and random stimuli, but curiously turned preferentially toward the latter. However, they showed no preference between biological and scrambled displays, results that match responses produced by vertebrates. Crucially, spiders turned toward the stimuli when these were only visible by the lateral eyes, evidence that this task may be eye specific. This represents the first demonstration of biological motion recognition in an invertebrate, posing crucial questions about the evolutionary history of this ability and complex visual processing in nonvertebrate systems.

For the vast majority of animals, determining if an object in the environment is another animal is crucial for survival. However, recognizing every single object individually quickly becomes computationally impractical. We thus expect evolution to favor simple but robust systems with general rules exploiting common characteristics that are able to cover most situations [1]. In most vertebrates, there are characteristic spatiotemporal relationships between different body parts while in motion—relationships imposed by the overall body plans of these animals.

funder had no role in study design, data collection and analysis, decision to publish, or preparation of the manuscript.

**Competing interests:** The authors have declared that no competing interests exist.

**Abbreviations:** ALE, anterior lateral eyes; AME, anterior medial eyes; GLMM, generalized linear mixed model; PLE, posterior lateral eyes; PME, posterior medial eyes.

Because the vertebrate body consists of linked rigid segments, movements between joints are accordingly semirigid—while some joints may seem independent from one another (e.g., wrist to knee) with their relative distances varying during locomotion, other joint pairs are not (e.g., wrist to elbow), with relative distances that are fixed. Thus, when observed visually, the movements of these animals result in a statistically identifiable idiosyncratic pattern. This is known as biological motion.

Indeed, even when presented with just 11 dots moving in correspondence with the position of the main joints of the human body, observers can correctly identify the presence of an agent [2–6]. Such dot-only stimuli (called point-light displays) are completely devoid of structural information, but retain motion-based information. The detection of these patterns in humans is not confined to stimuli depicting conspecifics, but enables the detection of many other animals [7]. Even "scrambled" displays, where dots maintain a semirigid relationship, are perceived as unknown, yet alive, entities [8]. Moreover, the discrimination of these patterns seems to be present at a very young age [9]. Studies on animal models also allowed to demonstrate the innateness of this ability [10,11]. Overall, studies in many other vertebrates [11–14], demonstrated that this visuo-cognitive strategy is widespread across vertebrates and is thus evolutionarily ancient. As such, biological motion detection is widely considered to be a key mechanism in enabling the detection of living animals within a visual scene [6,8,15]. That this system relies on motion [4,16] is important, as it has been proposed that motion-based visual cues can be extracted more quickly than many static cues, are less vulnerable to disruption, and are potentially governed by smaller networks of neurons [17].

In most arthropods, body plans and movements follow the same rules of semirigidity described above for vertebrates, since their exoskeleton is composed of connected, rigid components. Furthermore, like for many vertebrate species, the ability to visually differentiate moving animals from other visual stimuli is likely to provide a strong selective advantage. Because of these similarities in both form and function, we set out to explore whether biological motion cues might also be used by invertebrates, suggesting an even more widespread ability across animals.

We used jumping spiders (Salticidae) to address this question, as these animals are among the most visually adept of all arthropods, with vision playing a central role in a wide range of behaviors [18]. The visual system of these animals is also unique, with a total of 8 eyes and with different pairs thought to be specialized for different visual tasks [18]. The large anterior medial eyes (AME or primary eyes) are forward facing and characterized by a narrow visual field ($<5˚$) and high visual acuity [18] and are believed to play a central role in figure recognition [19,20]. The other 3 pairs are termed "secondary eyes." The anterior lateral eyes (ALE) have lower visual acuity but a much wider visual field ($\approx\pm50˚$) and are believed to be specialized for motion detection [21,22]. The posterior medial eyes (PME) are smaller and believed to be vestigial in many jumping spiders species [23]. The posterior lateral eyes (PLE) are characterized by a very wide visual field, and, together with the ALE, grant the spider a near 360˚ field of view. The function of the PLE seems to be similar to the ALE. Behavioral observations support the functional division of labor hypothesis: When a moving target is detected by the secondary eyes, the spider will pivot to face it directly and track it with the AMEs [21,24,25]. These rotational pivots are so rapid, precise, and robust that they are referred to as saccades [26], analogous to the eye movements of other animals, including humans. However, salticids do not turn toward all moving targets in the visual field, suggesting a selective attention–like process and/or that secondary eyes can inform target differentiation, such as the detection/discrimination of biological motion cues, rather than simply acting solely as motion detectors [27]. This idea is also supported by the fact that, neuroanatomically, the 2 sets of eyes (primary

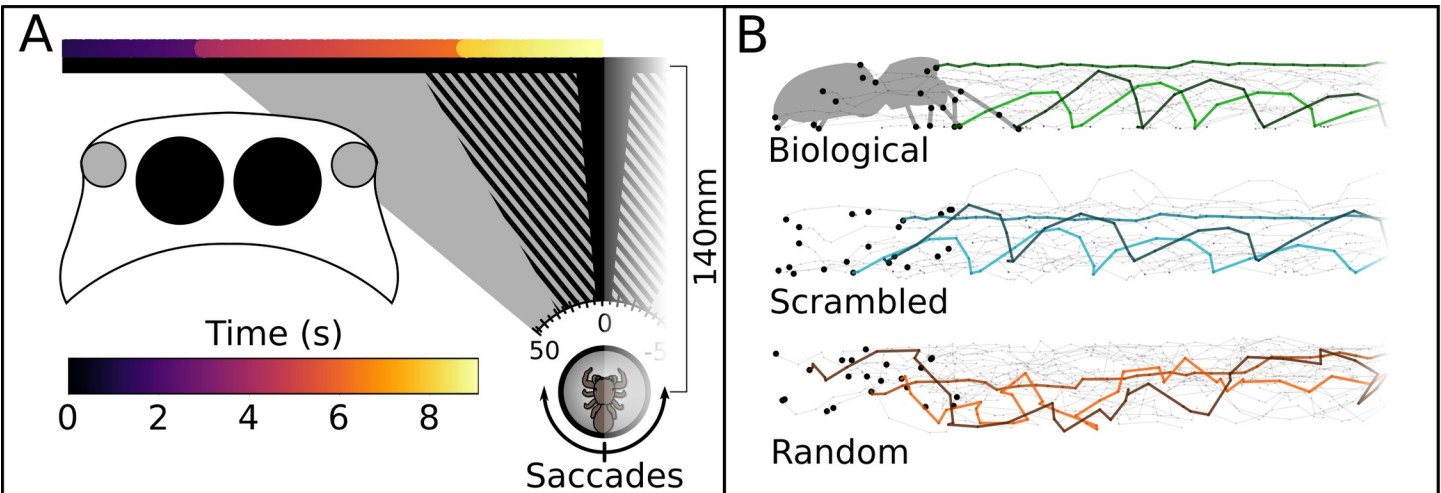

**Fig 1. Stimuli movement across the screen and of each point. (A)** Schematic representation of the setup, left half. The black horizontal line represents the computer screen. The colored line above represents the position of the stimulus across time (color scale). Note the sudden color changes where the stimulus paused for 1.5 seconds before starting to move again. Once at the center, the stimulus disappeared behind a white box. To turn toward any stimulus on the screen, the spider can produce saccades, by pivoting around its vertical axis (z-axis of the sphere). Cones drawn toward the screen: visual field of the different eyes. In black, the visual field of the AME. The striped area represents the extended visual field of the AME when moved. In gray, the visual field of the ALE. Note that the visual field of the remaining eyes covers all the rest of the screen. Large overlay schematic on left depicts a jumping spider cephalothorax, seen frontally with the AME in black, and the ALE in gray. **(B)** Point-light displays. On the left, the full set of dots for the first frame. For the biological, the silhouette has been superimposed on the dots, to show how they correlate with the spider's joints. For each stimulus, the paths for 3 points (the same 3 in all displays) is highlighted. Note how for the scrambled display, while the position of the points is different from the biological, the path that they follow is identical. ALE, anterior lateral eyes; AME, anterior medial eyes.

and secondary) possess completely separate early visual processing pathways [28], suggesting that there may be specialized computational tasks for which each network is responsible.

We tested *Menemerus semilimbatus* spiders in a forced-choice paradigm using point-light display stimuli. For a detailed description of the procedure, see Materials and methods section. Individuals were suspended above a polystyrene sphere so that their legs could contact it, while the sphere was supported and allowed to spin freely by a constant stream of compressed air from below. In spiders, the legs attach to the cephalothorax (the head)—thus, in this setup, the spider remained in a fixed position and maintained a fixed head orientation, but was able to move its legs freely, transferring its intended locomotor actions to the sphere. Stimuli were then presented on a computer monitor placed 14 cm in front of the spider (Fig 1A). Two stimuli were presented simultaneously, entering the monitor space from opposite sides and moved toward the center, where they disappeared. Using a video-based sphere tracking system (FicTrac) [29], we then measured how spiders moved the sphere in response to the stimuli presentation. As stated above, jumping spiders produce rotations or "saccades" with their body upon detection of a stimulus with the secondary eyes, turning rapidly about a single central axis (like a top or a military tank). Thus, the measured output was the rotational locomotor behavior—which of the 2 targets the spider chose to rotate toward.

We designed 5 different stimuli, paired in a total of 4 different conditions (Fig 1B; see Materials and methods section for details; videos of the stimuli are available in S1–S11 Videos). Stimuli were designed as spider versions of standard point-light display stimuli. The "biological motion" stimulus consisted of a point-light display following the motion of a walking spiders from a side-view perspective. In the "scrambled motion" stimulus, the starting point of each dot was randomized, but it then followed the same path as in the biological motion stimulus—a pattern perceived as biological in other systems [8]. The "random motion" stimulus consisted of the same number of dots constrained to the same overall area as the biological motion

stimulus, moving with the same velocities, but in randomized directions. A spider silhouette, based on the same video used to create the biological motion stimulus and thus following the same movements and an ellipse translating at the same speed of the other stimuli but without the finer-scale joints movement was also used. In conditions 1 and 2 (Fig 2C), the random motion stimulus was paired against the biological and the scrambled stimuli, respectively, testing responses to biological versus nonbiological motion. In condition 3, the biological stimulus was paired against the scrambled stimulus, thus presenting 2 semirigid stimuli. In condition 4, the spider silhouette was paired against the ellipse, a control condition to observe responses toward structurally detailed stimuli compared to moving stimuli of similar size but with minimal detail.

## Results

Only the main results are reported here. For the full analysis, see S1 Scripts. Firstly, we checked whether the observed rotations of the sphere did truly describe the reaction of the spider to the presence of any stimulus on screen. We found a strong correlation between absolute rotational speed and the stimuli angular location (both occupied the same angle, being symmetric to the center) (generalized linear mixed model [GLMM] analysis of deviance, chi-squared = 3330.8, $p$-value < 0.0001) across all experimental conditions (difference between conditions: chi-squared = 2.52, $p$-value = 0.47) (Fig 2A and 2B). We also observed a significant interaction between stimulus position and condition (chi-squared = 46.77, $p$-value < 0.0001): Specifically, we found this correlation to be stronger in the silhouette versus ellipse condition than in the other conditions (post hoc analysis, Tukey correction. Silhouette ellipse versus scrambled random: est. = 0.004, SE = 0.001, t = 4.73, $p$ < 0.0001 versus biological scrambled: est. = 0.005, SE = 0.001, t = 6.31, $p$ < 0.0001 versus biological random: est. = 0.004, SE = 0.001, t = 5.12, $p$ < 0.0001). This was likely due to the fact that the silhouette and the ellipse stimuli are composed of more black pixels, resulting in higher contrast between the stimuli and the white background than the point-light displays. This would have allowed those stimuli to be detected earlier, strengthening the correlation between rotational speed and stimulus position. All together, these observations suggest that the observed Z rotation of the sphere is indicative of saccades produced by the spiders toward the stimuli on screen.

Having established that, we proceeded by analyzing the average direction of the saccades. To do so, we multiplied by 1 Z rotation values in a direction congruent with the position of the more biological stimulus, while we multiplied them by −1 when congruent with the other. We then used this as the dependent variable to compute the spider's preference (Fig 2C). During stimulus presentation, the spiders turned more toward the ellipse over the silhouette (est. = −23.73, SE = 3.41, t = −6.96, $p$ < 0.0001), the random over the scrambled (est. = −15.1, SE = 3.41, t = −4.435, $p$ = 0.0001), the random over the biological (est. = −13.37, SE = 3.38, t = −3.96, $p$ < 0.0006), and showed no difference in the biological versus scrambled condition (est. = −1.04, SE = 3.3, t = −0.32, $p$ = 1). When considering the 30-second between-stimulus period instead, we observed no difference for all conditions (post hoc analysis, Tukey correction. Biological random: est. = 0.52, SE = 1.59, t = 0.32, $p$ = 1; biological scrambled: est. = 1.68, SE = 1.53, t = 1.1, $p$ = 0.922; scrambled random: est. = 2.46, SE = 1.68, t = 1.47, $p$ = 0.706; silhouette-ellipse: est. = 3.4, SE = 1.72, t = 1.97, $p$ = 0.328). Note that most saccades happens as soon as the stimuli start moving (Fig 2B). Among those sections, the higher frequency of rotation can be observed around 4.5 seconds from stimuli appearance (Fig 2B), when both are positioned at an angle of ≈±50˚ (Fig 1A). As stated in the introduction, this angular position likely aligns with the start of the ALE field of view.

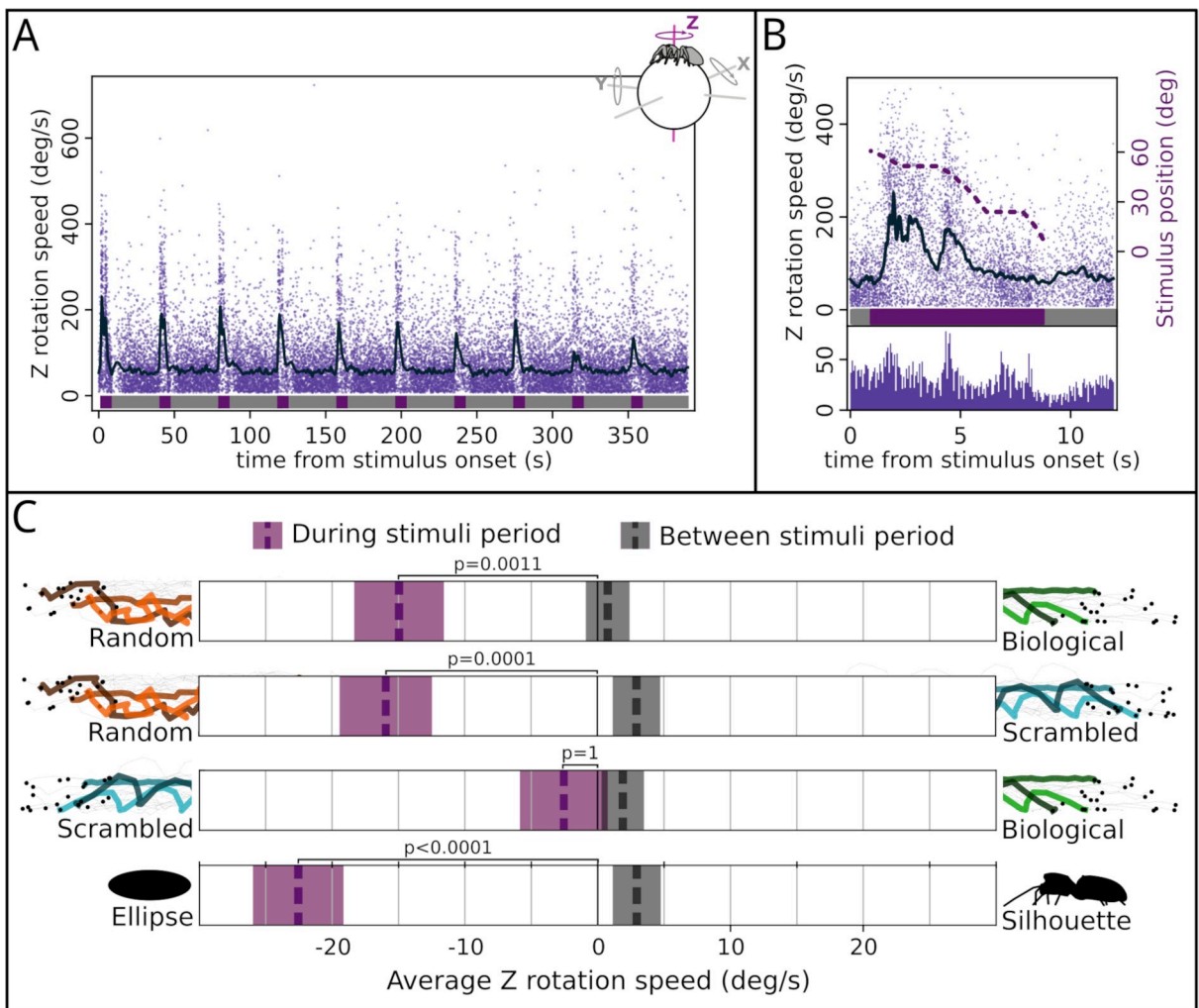

**Fig 2. Results. (A)** Recorded peak rotation speeds for each trial (every set of 10 stimuli pair presentations; $n^{trials} = 188$; $n^{spiders} = 60$). The x-axis represents time (in seconds) from the start of the experiment, and the y-axis represents rotational speed (in degrees per second). Individual dots represent peaks, while the dark line shows the running average. In the top-right corner, a schematic representation of the spider on top of the sphere, with the 3 possible axis of rotation. The data presented in this and the following graphs refer to rotation observed on the sphere's z-axis, which correspond to the spider's saccades. Below, colors show when the stimuli were onscreen (purple) or not present (gray). When stimuli are present, the average rotational speed increases, while when stimuli are not present, peak rotational speed remains low. This suggests that rotational speed is representative of saccades and that spiders are turning toward stimuli (see Materials and methods section). **(B)** Time-aligned responses across all stimuli. Upper portion: same as panel A, with addition of right y-axis indicating position of the stimuli on the screen (in degrees from the center, dashed purple line). Below portion: histogram of the peaks frequency. Peak frequency increases when the stimuli are moving and decreases when they are stationary (flat sections of the stimulus position line). **(C)** Stimulus preference for each condition. The x-axis represents the rotational speed. Positive or negative values correspond to a preference for the stimulus indicated. Purple boxes refer to sections when the stimuli were visible (during stimuli period), and gray boxes refer to sections when stimuli were not visible (between stimuli period). Dashed lines in the box represent the average, and shaded areas represent SE. For this analysis, every peak was either positive, when turning toward the stimulus depicted on the right of the graph (the one with biological characteristics), or negative, when turning toward the stimulus shown on the left. Thus, average speed is a proxy of relative preference, with 0 corresponding to no preference. For the biological versus random, scrambled versus random, and silhouette versus ellipse, the spiders preferred the less realistic stimulus. We observed no preference in the biological versus scrambled condition. Across all conditions, there was no preference during the 30-second between-stimuli period. Data underlying panels A and B are available in S1 Data. Panel C depicts result of the GLMM based on the same data. The analysis script is available in S2 Analysis. GLMM, generalized linear mixed model; SE, standard error.

## Discussion

These results clearly demonstrate the ability of jumping spiders to discriminate between biological motion cues. The results also show strong internal consistency, with preferences across all stimulus pairs for the less realistic stimuli, with the exception of the paired semirigid stimuli where no preference was observed. We initially found this "reverse choice" surprising, as we expected spiders to turn toward the stimulus with the highest probability of being a living organism, as has been previously found in other systems [9,11]. Even though the stimuli may have been perceived as predators or competitors, given their size, it seems unlikely that our result can be explained as an avoidance effect. Rather than turning toward the "less dangerous" stimulus under this perspective, we would have expected spiders to have maintained their attention on the "more dangerous" stimulus and possibly attempted to run away. However, orientation toward the less-biological stimulus appears consistent with the functional organization of the visual system of the jumping spiders. As described above, these animals produce saccades upon detection of a target with the secondary eyes, allowing further inspection with the AME. As per our initial hypothesis, the secondary eyes may be immediately able to decode motion-based information, enabling them to determine which of the stimuli requires more detailed investigation. In a forced-choice paradigm, it may be advantageous to focus the AME on the stimulus that cannot be decoded with the secondary eyes alone, particularly since the other target will still remain in the visual field of the secondary eyes following rotation. Moreover, it is crucial to consider that many of the species in which a preference for biologically moving stimuli has been observed are highly social. Indeed, it has been amply argued that an innate "animacy detection" system may exist because of the need of these animals to quickly aggregate with social companions [30]. In such species, it would be advantageous to show an early attentional preference to movements that are more likely to be generated from an agent. However, jumping spiders are not typically social; thus, we should not necessarily have the same expectation for stimulus preference. It is crucial to point out that motion cues in point-light displays can be used for much more than just "animacy detection." It has been shown that humans cannot only detect the presence or absence of a living being in the stimulus, but also much finer details, like, for example, the sex of the actor [31]. It has been suggested that, also, jumping spiders could detect and use more subtle motion cues to discriminate between different species to approach or avoid, like conspecifics or ants [32]. The lack of a preference between the scrambled and the biological displays in our experiment seems to point against the presence of such finer discrimination, but more direct inquiries are needed to provide a definitive answer.

In our setup, spiders could only observe the target with both AME and ALE eyes when the stimuli reached the center of the screen (see Fig 1A). Specifically, this opportunity for primary–secondary overlap occurs after ≈7s from the stimuli appearance, yet the majority of during-stimulus turns, and thus choices, occurred before this phased (see Fig 2B). That spiders demonstrate this preference even when targets can only be viewed by the secondary eyes is striking. Indeed, multiple studies have been carried out about the visual discrimination abilities of jumping spiders [33,34], and the secondary eyes have been found to possess a reasonably high spatial acuity [22,24,35]. However, even though it has been suggested that these eyes are capable of feats beyond mere motion detection [27], no study before had directly demonstrated that these eyes can solve a discrimination task without aid from the primary eyes. As point-light displays are designed to contain minimal visual detail, the discrimination operated by the secondary eyes must be based on motion.

Our findings provide evidence for the discrimination of point-light display biological motion stimuli [3] in an invertebrate. If this is evidence of convergent evolution or of deep

homology (e.g., presence in the shared common ancestor) still remains unclear. However, it is widely believed that the complexity of high-acuity vision in the vertebrates and invertebrates (e.g., vertebrates, insects, spiders, and mollusks) have evolved independently [36], suggesting that this is more likely a result of convergent evolution. Indeed, the detection of biological motion represents a solution or strategy to the problem of "animacy detection" [6,8], which could have been adopted by spiders and vertebrates alike. We suspect that, given this, future work in other lineages (e.g., insects and mollusks) may reveal similar results. Indeed, the use of motion perception by insects to extract other forms of information has been amply studied: For example, flies, bees, and wasps use optic flow information to calculate ground speed and thus navigate the environment [37], while locusts use of motion parallax to estimate distance [38]. These systems remain different from the detection of biological motion, as they rely on changes in the visual scene caused by one's own movement, while responses to dynamic targets still function when the observer is stationary.

The presence of a biological motion-based detection system in jumping spiders deepens questions regarding the evolutionary origins of this visual processing strategy and opens the possibility that such mechanisms might be widespread across the animal kingdom and not necessarily related to sociality.

## Materials and methods

### Subjects

We used 60 *M. semilimbatus* in the experiment, of which 31 were females, 10 were males, and 19 were juveniles. The spiders were collected in the wild, in the garden of Esapolis' living insect museum, Padua, Italy between June and August 2020. Only animals with a body length bigger than 7 mm were collected to guarantee the proper functioning of the methodology. Once caught, the spiders were maintained in clear plastic boxes measuring $80 \times 65 \times 155$ mm and immediately fed a small *Tenebrio molitor* to ensure a shared level of satiation before the test. The day after capture, a magnet was fixed to the head of each subject to allow us to constrain the animal on top of the treadmill for the duration of the experiment (see next paragraph). Subjects were first constrained between a sponge and a latex film. The latter presented a hole in correspondence with the location of the spider's head. Here, a $1 \times 1 \times 1$ mm neodymium magnet was applied using a UV activated resin. Each spider underwent its first test between 1 and 3 days after receiving the magnet. At the end of the full experiment, each spider was again constrained under the latex film to remove the magnet. Spiders were then freed in the same place they were captured. Magnets did not appear to negatively affect the animals during the short period in which they were housed in the lab, and spiders that had the magnet removed appeared to move and behave normally.

### Experimental apparatus

In order to ensure that spiders would always see the stimuli from the same position, orientation, distance, and, most importantly, both stimuli at the same time, we built a sphere-based treadmill, similar to that described in Moore and colleagues [29]. Similar procedures have been used in jumping spiders, and these animals appear to react to digital stimuli as if they were freely moving in the environment [22,25,39–41].

To construct the treadmill, a polystyrene sphere (diameter = 38 mm) was placed on top of a shallow concave holder. A second piece of plastic with a 24-mm hole in the center was placed on top of the sphere, leaving 1 mm of clearance where this plate met the sphere. In this setup, the sphere was free to rotate, but its overall position was fixed in place. The plastic holder was designed so that the side of the sphere was still mostly visible, with the plastic pieces covering

only around 30% of its surface, and the top portion of the sphere was left exposed so that the spider could contact/hold it. This plastic holder also contained 16, 1-mm diameter air inlets directed toward the sphere. When a compressed air source set to a constant pressure of 50 mbar was connected to the plastic holder, it produced an air cushion that the sphere floated on, providing almost frictionless rotational movement. The pressure was low enough not to cause the sphere to rotate by itself. The rotational inertia of the ball was also such that the spider could rotate it freely.

To initiate each trial, we used a stick with a magnet glued at one of its ends to pick up a spider. After adjusting the position/orientation of this stick, we positioned the spider on top of the sphere so that the center of the spider's cephalothorax was at the top of the sphere, its body aligned toward the screen, and at a height above the sphere that enabled a normal walking gait. In this setup, the subject could not move nor rotate, as its cephalothorax was fixed in place, but it would still behave as if it was able to move, acting directly on the sphere. By measuring sphere rotations, we could infer the spider's intended action.

To enable tracking of the sphere, we used acrylic paint to draw uneven shapes across its surface. From the side view, we recorded video of each experiment at 120 fps. By running the Fic-Trac software by Moore and colleagues [29] on these videos, we were able to extract the exact orientation of the sphere in each frame, expressed by its X (left-right), Y (fore-aft), and Z (rotational) rotational components, and, in turn, the rotational speed.

After every 4 trials, the system was disassembled to clean the sphere and the plastic pieces from any silk residue, which might interfere with sphere rotation.

## Stimuli of the experiment

All stimuli were based on a video recording of the side view of a *Salticus scenicus* walking from the right to the left of the frame. The video was captured with a Nikon D7200 DLSR camera (Nikon, USA), focused on a 5-cm wide runway recorded at 60 fps and 1080p. From the video, we extracted 72 frames, which contained a total of 8 full steps of the subject, with step defined as a movement of all 8 legs starting and terminating with each limb occupying the same position relative to each other. In each frame, we manually registered the position of the eyes, the pedicel, the spinneret, and for every leg, the end of the tarsus, the joint between the metatarsus and the tibia, and the joint between the patella and the femur. If any of the points were not visible in a specific frame (which did often happen especially for the right side legs, being covered by the body), the point position was registered as not available.

All stimuli were presented to the spider on a 1080p screen, with a pixel size of 0.248 mm, positioned 14 cm away from the center of the treadmill sphere. In each condition (except for the single-stimulus control trials), 2 stimuli were presented simultaneously, one entering from the left side and the other entering from the right, both moving toward the center, disappearing behind an ideal white screen in the exact center of the monitor. The presentation of the stimuli was controlled by a script written in python 3.8 [42], using the package PsychoPy [43]. The same script also controlled the camera recording of the treadmill, with the package OpenCV [44]. This way, stimulus presentation and camera recording were matched, in order to know the position of the sphere for every frame of the display. All the stimuli were presented on the screen at 30 fps. Videos of the stimuli are available in S1–S11 Videos.

**Biological motion.** For this point-light display stimuli, a black dot was presented at each anatomical location registered in the original video, rescaled in order for the stimulus to occupy a size of 170 × 60 px on the screen (the total width and height changed slightly following the extension and retraction of the spider legs). The starting position of the stimulus was set outside the screen, so that the picture would not be shown suddenly on the screen but

would instead appear to move in from outside. After every 72 frames (all the frames extracted from the original video), the stimulus would remain static for 45 frames, then loop over the original 72 frames, but, this time, translated in the X direction to center the first frame over the current position. This 45 frames pause was repeated another time and was followed by a third loop iteration, which terminated with the stimulus disappearing progressively behind a white box at the center of the screen. A full stimulus presentation, from start to end position, was a total of 270 frames, or 9 seconds at the display rate of 30 frames per second. The total time the stimulus was visible to the spider, thus excluding the time it took to appear from outside of the screen, was 8 seconds.

**Scrambled motion.** In the scrambled motion stimulus, the path of every dot was maintained, but their relative positions to one another in the first frame was randomized. In other species tested, scrambled motion stimuli are still perceived as biological, giving the impression to the observer of seeing an unknown living creature [8]. To generate this stimulus, we calculated for every point in each frame its distance to the position it occupied in the previous one, but while maintaining the center of the stimulus in a fixed position. Then, the position of every dot was randomized for the first frame. The position of the dots in the following frames would be determined by reapplying the distance and direction extracted from the original stimulus to the random position of the dot. We then reapplied the translation, moving all the dots by placing the center of the picture in correspondence with the center of the original stimulus. This way, the scrambled motion stimulus retained the same speed as the original, by globally translating the same amount, and each dot would follow the same path, but the relative position of the dots in relation to one another was eliminated. We created 4 different versions of this stimulus, with 4 different randomizations. For each trial, one of these 4 was chosen, so that each spider would never see the same randomization in different trials (for example, 2 different conditions present the scramble motion—biological versus scrambled and scrambled versus random—any one spider will see 2 different versions of the scrambled stimuli, 1 for each condition), and different spiders would see different versions for the same condition.

**Random motion.** In the random motion stimulus, the position of every dot in every frame was randomized, and according to the literature, this should not be perceived as biological [12]. To build this stimulus, we began with the initial, full stimulus used in the biological motion stimuli. We then followed the same procedure as for the scrambled motion but defined the position of the dots in frames 2 to 72 using the same distance from the previous frame as the biological motion stimulus, but in a random direction. Thus, the magnitude, or amount of motion, is maintained, but each dot takes a random path. To prevent dots from shifting too far from the center, we recalculated a new random direction if the resulting position of the dots exceeded the perimeter of an ideal rectangle of $170 \times 60$ px, exactly as the one enclosing all the dots in the biological motion stimulus. We created 4 different versions of this stimulus, with 4 different randomizations, following the same presentation as per scrambled motion.

**Silhouette.** This stimulus was intended as a control, to test whether spiders would show a preference in this setup at all. Previous experiments have shown that jumping spiders respond differently to different silhouettes [45], but we wanted to evaluate responses to stimulus presentation on our spherical treadmill. For this stimulus, we connected the dots from the biological motion stimulus belonging to the same leg, resulting in a thick black line representing the full leg in its movements. We also added black shapes to represent the spider cephalothorax and opisthosoma, based on the original walking spider video. One edge of each body segment shape was then centered on the picture and each shape tilted according to the eyes and spinneret dots. Thus, this stimulus moved in the same way as the biological motion stimuli, but provided a full silhouette of the target in terms of static appearance.

**Ellipse.** This shape matched exactly the silhouette in terms of the number of black pixels. For every frame, the center of the ellipse was placed to the coordinates of the center of the biological motion, and, in turn, all the other stimuli.

## Detailed procedure

Each spider underwent a total of 4 trials, 2 of which were administered on a given day, and the other 2 were administered after a 3-day break. Every trial started with 5 minutes of habituation, in which no stimulus was shown, just a white screen. This phase was included in the experiment to give the animals enough time to decrease their level of arousal from the manipulation before the experiment started and revert to normal behavior. After habituation, the first presentation started, where 2 stimuli would appear, one from the left and one from the right, move toward the center of the screen, and then disappear. As described, the stimuli stop twice, for around 1.5 seconds, before reaching the center. After the stimulus disappearance, a 30-second pause, with no presentation, followed. This pattern of presentation pause was repeated another 9 times, for a total of 10 stimuli presentation. Thus, a full trial lasted 11 minutes and 30 seconds. In a given trial, the same stimulus pair was shown for all the 10 stimuli presentation, and the starting position of each stimulus (left or right) would follow a semi-random pattern (left, right, left, right, left, left, right, right, left, right or vice versa for the other stimulus). This way, each trial was part of a single condition. Each spider was subjected to a different condition (namely, a specific stimuli pair) in every trial and was therefore presented with all experimental conditions. The order of the trials was randomized for each spider.

After the end of the experiment, we added a fifth condition: ellipse versus nothing. Only 1 subject was used to test it, as it was only intended to retrieve a baseline measure of preference, to inform the interpretation of the data of the other conditions.

## Data analysis

The sphere tracking software FicTrac [29] provided, among other measures, the radians per frame that the sphere rotated around its x-, y-, and z-axis, for each video frame (Fig 2A). A rotation around each of these axes represented a different kind of motion of the spider:

- x-axis: Parallel to the screen, the rotation of the sphere around this axis would occur when the subject would move forward or backwards. From the perspective of the side-view camera that was used, a counterclockwise rotation would happen when the spider moved forward and vice versa.

- y-axis: Perpendicular to the screen, such that rotation around this axis would happen if the spider moved directly sideways (left or right).

- z-axis: Vertically, through the spider, such that rotation would be registered when the spider rotated around its center. A turn to the left of the spider would correspond to a clockwise rotation and vice versa.

Other types of movements (e.g., the spider moving diagonally) would result in a combination of the 3 rotations. Before further analysis, the values of each axis were smoothed with a Butterworth filter (order = 3 and critical frequency = 0.5) from the python package SciPy [46]. What we were interested in were the turns toward the left and toward the right, which would correspond with z-axis rotation peaks in the absence of X and Y components. Due to noise in the system and to individual differences of the spiders and videos, movements typically contained rotational velocities across all 3 axes. To reduce the impact of these cross-axes effects, for every frame, we subtracted from the absolute value of Z the absolute value of X and Y, with

0 as the minimum allowable resulting value. Then, the original sign of Z component (+ or −) was reapplied. This way, Z peaks with low concurrent X and Y components remained high, while highly coupled rotations would disappear. Thus, saccadic movements (large Z rotations and low X and Y components) could be clearly separated from non-saccadic movements such as walking forwards (large X and low Y and Z) or curved walking (large X, large Y, and medium Z).

To analyze the impact of stimuli on saccadic turns, we focused on the stimuli presentation part of the trials, including the 30-second pauses between stimuli pair presentations, and changed the sign of each rotation according to the position of the expected preferred stimulus: Instead of positive numbers meaning left and negative meaning right, positive numbers represented rotations toward the more biologically realistic stimulus and vice versa. For example, in condition biological motion versus scrambled motion, biological motion was coded as positive. Note that because of this conversion, the scrambled motion stimulus was negative in one condition (biological versus scrambled) but positive in another (scrambled versus random). The same coding was maintained for the pause section following each stimulus, with rotation toward the side in which the correct stimulus was previously present being positive. We then isolated peaks higher than 0.001 rad/sec for both the positive and the negative space and used these peak heights and times in our analysis.

With this coding, a saccadic preference for the more realistic stimulus (i.e., more turns toward it) would result in a positive peak average, saccadic preference for the less realistic stimulus will result in a negative average, and an average of 0 would signify no saccadic preference. Indeed, we expected to observe no preference for the between-stimulus sections, regardless of condition, given the a priori expectation that rotational movements in the absence of stimuli should be directed randomly.

## Statistical analysis

All data and graphs were prepared using python 3.8 [42] with the libraries pandas [47,48], NumPy [49,50], SciPy [46], and Matplotlib [51]. Analyses were carried out with R 4.0.2 [52], using the packages glmmTMB [53], car [54], emmeans [55], DHARMa [56], and readODS [57]. For the full script, see Supporting information text. The full database is available as a Supporting information. As suggested by Forstmeier and Schielzeth [58], we included in the models only factors that we had an a priori reason for including. First, we calculated a model with the absolute (without the sign coding the direction) peak height as a dependent variable and the angular position of the stimulus on the screen as a predictor. We employed a generalized linear mixed-effect model, with subject as a random effect. This analysis was performed to check whether extracted peak height represent rotational speed of the spider. Indeed, we know from previous literature [22,25] that when moving on similar spherical treadmills, the rotation caused by the detection of a target (saccadic turns) performed by spiders match very closely the actual angular position of the target. Moreover, these saccadic rotations tend to take similar amounts of time, resulting in faster rotation when the target angle is higher. Subsequently, we calculated a second model, using peak height (with positive values when its direction was congruent with the more realistic target and vice versa) and the dependent variable, condition, and stimulus presence (visible or not visible) as predictors. Here, we expected the average peak height to be indistinguishable from 0 when the stimulus is not visible (as saccadic rotations should be less frequent, and even if they did appear, they should be toward a random direction) and to be higher than 0 for conditions biological versus random, scrambled versus random, and silhouette versus ellipse, signifying a preference for the biologically structured stimulus.

### Methods validation results

To check whether our scoring based on the average Z rotation was indeed an effective proxy for preference, in condition 5, we presented spiders with a choice between a moving ellipse and nothing on the other side. As expected, we observed a clear preference for the ellipse position when the stimulus was on screen (post hoc analysis, Tukey correction est. = 107.0, SE = 12, t = 8.89, $p < 0.0001$) and no preference in the 30-second between-stimulus time (est. = −24.5, SE = 13.9, t = 1.76, $p = 0.155$), supporting the utilized score as representative of spiders saccadic preference.

## Supporting information

**S1 Data. Data used in the analysis.** The first sheet contains metadata. The second sheet contains all the data for the experiment with the X, Y filtering on Z applied. The third sheet contains the control with the ellipse put in comparison with no stimulus. The fourth sheet is the same as the second, but without the filtering applied.
(XLSX)

**S1 Script. R and python scripts.** This compressed folder contains 2 file. The first file (Analysis) contains the full R script of the data analysis. The second one (DataFiltering) contains the python script used to subtract X and Y value for the Z, as described in the Materials and methods section, and to find peaks.
(ZIP)

**S1 Video. Biological motion point-light display.** The size, speed, and positions are true to the experiment. This example moves left to right, for right to left version the video was mirrored. A second stimulus, depending on the condition, would follow the same path, just mirrored.
(MP4)

**S2 Video. Spider silhouette.** The size, speed, and positions are true to the experiment. This example moves left to right, for right to left version the video was mirrored. In the experiment, this stimulus was always paired with the ellipse, which followed the same path, just mirrored.
(MP4)

**S3 Video. Ellipse.** The size, speed, and positions are true to the experiment. This example moves left to right, for right to left version the video was mirrored. In the experiment, this stimulus was always paired with the silhouette, which followed the same path, just mirrored.
(MP4)

**S4 Video. Scrambled motion point-light display, first randomization.** The size, speed, and positions are true to the experiment. This example moves left to right, for right to left version the video was mirrored. A second stimulus, depending on the condition, would follow the same path, just mirrored.
(MP4)

**S5 Video. Scrambled motion point-light display, second randomization.** The size, speed, and positions are true to the experiment. This example moves left to right, for right to left version the video was mirrored. A second stimulus, depending on the condition, would follow the same path, just mirrored.
(MP4)

**S6 Video. Scrambled motion point-light display, third randomization.** The size, speed, and positions are true to the experiment. This example moves left to right, for right to left version the video was mirrored. A second stimulus, depending on the condition, would follow the

same path, just mirrored.
(MP4)

**S7 Video. Scrambled motion point-light display, fourth randomization.** The size, speed, and positions are true to the experiment. This example moves left to right, for right to left version the video was mirrored. A second stimulus, depending on the condition, would follow the same path, just mirrored.
(MP4)

**S8 Video. Random motion point-light display, first randomization.** The size, speed, and positions are true to the experiment. This example moves left to right, for right to left version the video was mirrored. A second stimulus, depending on the condition, would follow the same path, just mirrored.
(MP4)

**S9 Video. Random motion point-light display, second randomization.** The size, speed, and positions are true to the experiment. This example moves left to right, for right to left version the video was mirrored. A second stimulus, depending on the condition, would follow the same path, just mirrored.
(MP4)

**S10 Video. Random motion point-light display, third randomization.** The size, speed, and positions are true to the experiment. This example moves left to right, for right to left version the video was mirrored. A second stimulus, depending on the condition, would follow the same path, just mirrored.
(MP4)

**S11 Video. Random motion point-light display, fourth randomization.** The size, speed, and positions are true to the experiment. This example moves left to right, for right to left version the video was mirrored. A second stimulus, depending on the condition, would follow the same path, just mirrored.
(MP4)

**S12 Video. Example of a single-stimulus pair presentation for the condition "silhouette versus ellipse".** Initially, the spider is in a normal locomotion phase. Around second 18 of the video, the first saccade can be appreciated, specifically toward the right side of the screen where the ellipse stimulus appeared (which will enter the visual field of the camera shortly). Then, some more saccades follow.
(MP4)

## Acknowledgments

We thank Esapolis' Living Insect Museum of the Padua Province, Padua, Italy and its director Enzo Moretto for providing the extra facilities unexpectedly needed due to the Coronavirus Disease 2019 (COVID-19) pandemic. We also thank Maria Loconsole and Lucia Regolin for suggestions regarding the manuscript.

## Author Contributions

**Conceptualization:** Massimo De Agrò, Daniela C. Rößler, Kris Kim, Paul S. Shamble.

**Data curation:** Massimo De Agrò.

**Formal analysis:** Massimo De Agrò.

**Funding acquisition:** Paul S. Shamble.

**Investigation:** Massimo De Agrò.

**Methodology:** Massimo De Agrò, Kris Kim.

**Project administration:** Paul S. Shamble.

**Resources:** Paul S. Shamble.

**Software:** Massimo De Agrò.

**Supervision:** Paul S. Shamble.

**Writing – original draft:** Massimo De Agrò.

**Writing – review & editing:** Massimo De Agrò, Daniela C. Rößler, Kris Kim, Paul S. Shamble.

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
