## [Editor Report · Decision Letter 0]

22 Mar 2021

Dear Dr De Agrò, 

Thank you for submitting your manuscript entitled "Perception of biological motion in a jumping spider" for consideration as a Short Report by PLOS Biology.

Your manuscript has now been evaluated by the PLOS Biology editorial staff, as well as by an academic editor with relevant expertise, and I'm writing to let you know that we would like to send your submission out for external peer review.

Please re-submit your manuscript within two working days, i.e. by Mar 24 2021 11:59PM.

Kind regards,

Roli Roberts

Senior Editor

PLOS Biology

---

## [Decision Letter · Decision Letter 1]

7 May 2021

Dear Dr De Agrò,

Thank you very much for submitting your manuscript "Perception of biological motion in a jumping spider" for consideration as a Short Report at PLOS Biology. Your manuscript has been evaluated by the PLOS Biology editors, an Academic Editor with relevant expertise, and by three independent reviewers.

You’ll see that all three reviewers are broadly positive about your manuscript. Reviewer #2 has a request for an additional analysis, and all three reviewers raise textual and presentational issues, but our impression is that there is not much work to do. We do note that, contrary to what rev #3 says, you already provide some supplementary videos.

In light of the reviews (below), we are pleased to offer you the opportunity to address the comments from the reviewers in a revised version that we anticipate should not take you very long. We will then assess your revised manuscript and your response to the reviewers' comments and we may consult the reviewers again.

We expect to receive your revised manuscript within 1 month.

**IMPORTANT - SUBMITTING YOUR REVISION**

*Resubmission Checklist*

*Published Peer Review*

*PLOS Data Policy*

*Blot and Gel Data Policy*

Sincerely,

Roli Roberts

Roland Roberts

Senior Editor

PLOS Biology

rroberts@plos.org

REVIEWERS' COMMENTS:

Reviewer #1:

The paper reports the first evidence of recognition of biological motion in an invertebrate species, a finding that I believe deserves to be published. I have only minor requests for changes.

p. 2 second para. line 10: This is incorrect of course. Human newborns cannot have been completely prevented the possibility to see motion patterns. Direct evidence is possible only in controlled conditions in animal models (e.g. the Plos Biology 2005 paper quoted below on newly-hatched chicks does provide such an evidence). This part should be better formulated. Please also consider on the same topic Gravity bias in the interpretation of biological motion by inexperienced chicks. Current Biology (2006), 16: 279-280.

p. 2 second para. line 15: On the 'animacy detector' see also Vallortigara, G. (2012). Aristotle and the chicken: Animacy and the origins of beliefs. In "The Theory of Evolution and its Impact" (A. Fasolo, ed.), pp. 189-200, Springer, New York.

p. 3 second para. the part starting from "Neuroanatomically..." This seems to be of little importance with respect to the rest of the paper and should be reduced or better omitted altogether.

p. 6 line 5 and line 11: change "agent detection" with "animacy detection" (recognition of agency is different than recognition of 'animacy')

p. 6 second para., the last three lines "How this might be implemented..." should be omitted.

p. 7 line 2: "animacy detection" not agent

Reviewer #2:

This manuscript details the responses of salticid spiders toward point light stimuli with different degrees of biological relevance (or 'realism') with respect to the movement depicted by the dots. This is a novel and interesting method and investigative tool to use on this group of animals. The results were somewhat surprising, and the authors have done a reasonable job of interpreting these results. I have mostly fairly minor comments, although there are some fairly glaring mistakes with the use of terminology of the eyes, and with respect to the descriptions of the fields of view (fov) of the different pairs of eyes. These can be easily rectified.

My principal comment is that, as you tested male and female spiders, and the sexes are known to be somewhat different in their contrast (and other) visual thresholds (Zurek work), it would have been interesting to see a comparison between males and females in this piece of work. I know this would require additional analysis, but I do think that this would make the current manuscript considerably stronger, both in content, and in its ability to make predictions about the visual system of this group of animals. Given that you have the rotational paths of each individual, and you know M/F, it should not be too hard?

All other comments and notation is made with respect to pages on downloaded pdf, as there are no line numbers.

On page 9, I am unclear how in joints "relative distances varying across time,". Do you mean phylogeny? I think you don't mean that - perhaps delete "across time", or of not clarify what you mean.

P10. Eye structure- this is incorrect. Note that the PME have a very narrow field of view, if any at all, in most species (where this pair of eyes is typically vestigial). The ALE and PLE have wide fields of view. See Land, M. F. (1985a). "Fields of view of the eyes of primitive jumping spiders." J. exp. Biol. 119: 381-384

Fig 2C. I am having some trouble working out the dark lines (mean) versus shaded area (SEM) - partly because there is coloured shading representing stimulus ON/OFF screen. Can a dashed line be used for the mean and shading for the SEM?

End of results. For people who understand salticid eyes and the set-up, it is not especially surprising that stimuli elicited a response at about 4.5 s, when they were within the fov of the ALE. However, for others, this sentence might appear a bit 'random'. I would make this clear.

P 12. "detection of a target with the lateral eyes" - you should be consistent. This is not technically correct. You should be referring here to the secondary eyes unless you are discussing solely your results in the narrower context, in which case this should be made clear and it should be stated that this is the ALE you are referring to and specifically with respect to this experiment.

P. 13, beginning. "…AME on the stimulus which cannot be decoded with the lateral eyes alone, particularly since the other target will still remain in the visual field of the secondary eyes following rotation." The work in Zurek, D. B., et al. (2010). "The role of the anterior lateral eyes in the vision-based behaviour of jumping spiders." Journal of Experimental Biology 213(14): 2372-2378, seems especially pertinent here, as it clearly demonstrated the spatial ability of the ALE, which is something that the authors do not seem to be aware of. This reference is also pertinent to your later statement: "That spiders demonstrate this preference even when targets can only be viewed by the secondary eyes is striking". The point made later, about discriminations being made on motion is very valid, and an interesting one.

P. 13. Some salticid species are, in fact, somewhat social. I would add the caveat "jumping spiders are not TYPICALLY social"

Very minor

Sometimes "cm" is written directly after the number (no space), and other times there is a space.

Reviewer #3:

In this interesting research paper, De Agrò and colleagues investigate the responses of jumping spiders to visual displays presented on a computer screen. The Authors focus on response to biological motion vs other manipulations of visual stimuli implemented using point light displays and silhouettes. Similarly to other studies conducted in vertebrate species, the Authors used pairs of stimuli including point displays of biological motion (that move semi-rigidly following the movement of a spider) vs scrambled motion vs random motion. They also tested a silhouette of a spider vs an ellipse. All stimuli translated horizontally, while joints/silhouettes were manipulated in different ways. 

The results solidly support the idea that in jumping spiders biological motion produces different behavioural responses (eg saccades) compared to random motion and scrambled motion. Although the visual representation of results is less than straightforward, the results appear robust and novel.

It would be interesting to know whether all spiders had similar responses or whether the effect is driven by few individuals.

Can the Authors clarify how many saccades were present in each condition? 

The Authors commented on the "unexpected" result of a preference of spiders for orienting towards the less naturalistic stimulus. Some potential interpretations have not been mentioned. For instance: Can this be due to the fact that the more naturalistic stimuli are considered as potential competitors for catching a potential prey? Is it possible that more realistic stimuli are considered potential predators to flee away from? These two options are basically prey capture and predator avoidance or competitor avoidance. Other solitary species appear to actively move away from conspecifics (e.g. tortoise hatchlings), so there are several possible reasons behind this outcome. 

This study leads to further questions on whether the size of objects and type of biological motion can influence the direction of preference.

I would like to ask the Authors to consider to use a more straightforward visual representation to help the reader. For instance, can they use legends for the different coloured lines? Is it possible make a connection between the unit of measurement to the variable of interest? (e.g. rotation toward the object?). 

In the Data analyses part I was initially confused by the X vs Y axis of the sphere. Would it be worth to add a visual representation of the sphere and its axis?

This paper would also benefit from a video of the subjects performing the task.

---

## [Editor Report · Decision Letter 2]

29 May 2021

Dear Dr De Agrò,

Thank you for submitting your revised Short Reports entitled "Perception of biological motion in a jumping spider" for publication in PLOS Biology. The Academic Editor and I have now assessed your responses and revisions. 

Based on his assessment, we will probably accept this manuscript for publication, provided you satisfactorily address the following points:

a) Please could you make a slight change to the title to make it most explicit for our broader readership? We suggest "Perception of biological motion by jumping spiders"

b) Please supply a blurb, according to the instructions in the submission form.

c) Many thanks for supplying the data underlying the Figures. Please could you rename the data file from “supp-2.xlsx” to “S1_Data.xlsx” and cite it in the legends to Figs 1 and 2 (e.g. "The data underlying this Figure can be found in S1 Data")?

We expect to receive your revised manuscript within two weeks. 

*Published Peer Review History*

*Early Version*

Sincerely,

Roli Roberts

Senior Editor,

rroberts@plos.org,

PLOS Biology

DATA NOT SHOWN?

---

## [Editor Report · Decision Letter 3]

11 Jun 2021

Dear Dr De Agrò,

On behalf of my colleagues and the Academic Editor, Lars Chittka, I'm pleased to say that we can in principle offer to publish your Short Reports "Perception of biological motion by jumping spiders" in PLOS Biology, provided you address any remaining formatting and reporting issues. These will be detailed in an email that will follow this letter and that you will usually receive within 2-3 business days, during which time no action is required from you. Please note that we will not be able to formally accept your manuscript and schedule it for publication until you have made the required changes.

PRESS: We frequently collaborate with press offices. If your institution or institutions have a press office, please notify them about your upcoming paper at this point, to enable them to help maximise its impact. If the press office is planning to promote your findings, we would be grateful if they could coordinate with biologypress@plos.org. If you have not yet opted out of the early version process, we ask that you notify us immediately of any press plans so that we may do so on your behalf.

Sincerely,

Roli Roberts

Roland G Roberts, PhD 

Senior Editor 

PLOS Biology

rroberts@plos.org